# Effect of Oxidized β-Carotene on Swine Growth Performance under Commercial Production Conditions in Vietnam

**DOI:** 10.3390/ani12223200

**Published:** 2022-11-18

**Authors:** La Van Kinh, William W. Riley, James G. Nickerson, La Thi Thanh Huyen, Graham W. Burton

**Affiliations:** 1Institute of Animal Sciences for Southern Vietnam, Di An 75309, Binh Duong, Vietnam; 2International School, Jinan University, Guangzhou 510632, China; 3Avivagen, Inc., Ottawa, ON K1A 0R6, Canada

**Keywords:** antibiotic alternative, diarrhea, health and growth performance, oxidized β-carotene

## Abstract

**Simple Summary:**

It is important to find suitable non-antibiotic alternatives for livestock production. The effects of oxidized β-carotene (OxBC) versus antibiotic growth promoters upon the growth performance of swine were determined through a full 140-day growth cycle in 500 barrows and gilts under commercial production conditions in Vietnam. There were five dietary treatment groups: Control basal diet with no antibiotics or OxBC; Basal diet with antibiotics and no OxBC; Basal diet supplemented with 2, 4, or 8 mg OxBC/kg diet and no antibiotics. OxBC and antibiotics each improved growth rate, feed efficiency, and body weight compared to the unsupplemented control. However, animals receiving 4 and 8 mg/kg OxBC performed better than animals on antibiotics. In Starter pigs (Days 1–28), OxBC reduced the occurrence of diarrhea dose-dependently (2, 4, and 8 mg/kg) and more so than antibiotics. These findings show that oxidized β-carotene can facilitate swine growth and health in the absence of in-feed antibiotics.

**Abstract:**

The effects of oxidized β-carotene (OxBC) were determined upon the growth performance of swine through their full growth cycle under commercial production conditions in Vietnam. Five hundred 28-day-old-weaned barrows and gilts were used in a 140-day complete wean-to-finish feeding trial. Animals were randomized by weight, and each pen contained 20 pigs with the same ratio of barrows to gilts. There were five dietary treatment groups with five replicate pens per treatment: Control basal diet, no antibiotics or OxBC; Basal diet with antibiotics; no OxBC; Basal diet supplemented with 2, 4, or 8 mg OxBC/kg of diet, no antibiotics. Growth performance parameters were calculated for each production stage (Starter: Days 1–28, Grower: Days 29–84, Finisher: Days 85–140) and for the overall study (Days 1–140). OxBC and antibiotics each improved growth rate, feed efficiency, and body weight compared to the unsupplemented control (*p* < 0.001). Animals receiving 4 and 8 mg/kg OxBC performed better than animals on antibiotics (*p* < 0.001). In Starter pigs, OxBC reduced the occurrence of diarrhea dose-dependently (2, 4, and 8 mg/kg) and more so than did antibiotics (*p* < 0.001). These findings support the concept that oxidized β-carotene can facilitate swine growth and health in the absence of in-feed antibiotics.

## 1. Introduction

It has been reported that the spontaneous non-enzymatic oxidation of β-carotene produces an oxygen-rich copolymer compound predominantly with immunomodulatory properties [1,2]. Fully oxidized β-carotene, termed OxBC, is rich in copolymer compounds and exerts its actions on the immune system through pathways that are distinct from either vitamin A or intact β-carotene, which are both absent. Johnston et al. proposed that copolymer compounds are, in fact, the agents responsible for many of the provitamin A-independent activities of β-carotene and other carotenoids [2].

OxBC exhibits dual immunological activities relating to (a) enhanced innate immune detection and response to pathogens [2] and (b) anti-inflammatory/pro-resolution that limits the extent of over-zealous immune responses and reduces the level of background inflammation [3,4].

The utility of OxBC as a feed additive has been demonstrated in studies with piglets [5], sows [4], broiler chickens [6], and dairy cattle [7]. In piglets, dietary supplementation with OxBC improved growth performance and prevented the vaccine-induced growth lag associated with the PRRS (porcine reproductive and respiratory syndrome) vaccination [5]. In sows, supplementation with OxBC, beginning at late gestation and continuing through lactation, resulted in reduced proinflammatory cytokine levels in colostrum and milk concurrent with increased colostral and milk immunoglobulin levels [4]. In broilers, dietary supplementation with OxBC reduced the level of pathogen (*Clostridium perfringens*) recovered from the gut and protected against the reduction in growth performance associated with experimental induction of subclinical necrotic enteritis [6]. Dietary OxBC resulted in increased cure-rates in dairy cattle with subclinical intramammary infections [7].

Several authors have proposed that the search for suitable alternatives to antibiotic growth promoters should focus on substances that achieve effects similar to the antibiotics they are intended to replace, namely, a reduction in both the bacterial load and inflammation [8,9,10,11]. The results from trials with pigs, poultry, and bovine dairy cows highlight the utility of OxBC in achieving both outcomes. The benefits observed with piglets and gestating/lactating sows likely are consistent with the anti-inflammatory actions of OxBC, whereas the reduction in *C. perfringens* in the poultry study and the elimination of the intramammary infections in lactating dairy cows support the benefits of innate immune priming. Note that OxBC has no direct anti-microbial effect [12] and that the reduction or elimination of bacterial pathogens in the poultry and dairy studies reflect actions on the host’s immune system, which is better able to detect and respond to the presence of pathogens [2].

In Vietnam, swine production has been dependent upon the use of multiple antibiotic growth promoters due to the heavy disease pressure in the country, caused in large part by the density of production. However, the Government of Vietnam has been eliminating access to selected antibiotics, particularly those that are important to human medicine. In this regard, the polymyxin antibiotic colistin, which is used to treat gram-negative bacterial infections, often as a last resort, has already been banned for use for prophylactic veterinary purposes in Vietnam. In addition, at the end of 2022, all antimicrobial growth promoters (AGPs) except halquinol will be banned except for treatment use, and halquinol itself will be eliminated as an AGP in 2025. There is a clearly established need for alternative solutions for the swine industry.

It is hypothesized that OxBC added at part-per-million levels to the diet of production pigs will support optimal health and performance in a manner similar to AGPs. The study objective was to determine the effects of OxBC on the performance of swine through their full growth period under commercial production conditions in Vietnam. The effect of OxBC on growth performance and the general health of male and female pigs over an entire 140-day growth cycle was evaluated relative to animals receiving a non-medicated control diet as well as a medicated diet containing an AGP regimen typical of commercial diets in Vietnam.

## 2. Materials and Methods

### 2.1. Animal Care

All animal care procedures followed those approved by the Animal Care and Use Committee of the Institute of Animal Sciences for Southern Vietnam. These procedures were established in accordance with Vietnam’s Law on Animal Husbandry, which covers the humane treatment of livestock during transport, slaughter, scientific research, and other activities. The study was issued approval number 121/QD-PVCNNB-KHHTQT by the Institute of Animal Sciences for Southern Vietnam.

The animals were monitored daily for signs of illness by farm staff. Suspect pigs were examined by a veterinarian who determined if they could remain in the trial or if they were to be removed due to observable pathologies. Mortalities were immediately removed from the pens and recorded.

### 2.2. Animals, Diets, and Experimental Design

The trial was carried out at a commercial farm, Thai My Pig farm, Thai My commune, in the Cu Chi district, Ho Chi Minh City. Five hundred weaned barrows and gilts (28 days of age) were used in a 140-day complete wean-to-finish feeding trial. The animals were reared on site and were the progeny of Landrace x Yorkshire sows and Duroc boars. The herd was asymptomatic for the porcine reproductive and respiratory syndrome (PRRS) and vaccinated against hog cholera (Coglapest™, Ceva-Phylaxia Veterinary Biologicals Co. Ltd., Budapest, Hungary), foot-and-mouth disease (Biotaftogen™, Biogénesis Bagó, Garin, Argentina), and PRRS (Tai Xanh™, Navetco, Ho Chi Minh City, Vietnam).

Animals were randomized by weight, and each pen contained 20 pigs with the same ratio of barrows to gilts. The trial was comprised of five dietary treatment groups with five replicate pens per treatment, as follows: Control (basal diet with no antibiotics or OxBC), AB (basal diet with antibiotics; no OxBC), and OxBC (basal diet supplemented with 2, 4 or 8 ppm (mg/kg) OxBC; no antibiotics). The compositions of the basal diets for the Starter, Grower, and Finisher phases of the study are shown in Table 1. No in-feed or water-administered medications or feed additives were employed except for the antibiotics used in the AB group and the OxBC used in the three treatment groups. OxBC was provided in-feed as the OxC-beta™ Livestock 10% commercial premix product (Avivagen Inc., Ottawa, Canada). Chlortetracycline and colistin sulfate were purchased from Zhumadian Huazhong Chia Tai Co. Ltd., Zhumadian, China and Lifecome Biochemistry Co. Ltd., Nanping, China, respectively.

For logistical and space purposes, the trial was conducted in a time-replicated fashion with replicates one and two for all five treatment groups beginning on Day 1, replicates three and four for all treatment groups beginning 28 days later, and replicate five for all treatment groups beginning 56 days after the first cohort.

Each pen was equipped with a dry hopper feeder equipped with six feed access holes, each measuring 13 cm × 13 cm, aligned parallel to the pen front. Feeders were located at the front of the pen, and a water drinker was located at the back of each pen. The pigs had ad libitum access to feed and water throughout the trial.

Pigs were individually weighed at the start of the study (Day 1) and at Days 28 (~20 kg), 84 (~50 kg), and 140 (~100 kg). All administered feed was weighed daily, feeders were emptied weekly, and the remaining feed was weighed. Growth performance parameters were calculated for each stage of the production cycle (Starter: Days 1–28, Grower: Days 29–84, and Finisher: Days 85–140), as well as for the overall study period (Days 1–140). The general health of the animals was assessed as the percent diarrhea incidence for each treatment group during the Starter, Grower, and Finisher phases. Any mortalities that occurred during the study were also recorded.

### 2.3. Statistical Analysis

Data for the experiment were analyzed by ANOVA as a randomized complete block design using the PROC MIXED procedure of SAS (Version 9.1, SAS Inst. Inc., Cary, NC, USA). Data were averaged across pigs in each pen where the pen was the experimental unit. The linear model used included block (body weight), treatment, and block × treatment (random error) to evaluate differences among the five treatments. Polynomial contrasts (linear and quadratic) were used to determine the dose-dependent effect of OxBC feeding. Results were reported as least square means and standard deviations, and differences were considered statistically significant at *p* ≤ 0.05, with tendencies identified when probabilities were greater than 0.05 and less than or equal to 0.10 (0.05 < *p* ≤ 0.10).

## 3. Results

Dietary supplementation with OxBC at 2, 4, or 8 ppm led to significant quadratic increases in final body weight and overall (Study Days 1–140) average daily weight gain (ADG) relative to both the Control and the AB groups (Table 2). Notably, the final body weight for the AB group, as measured on Day 140, was higher than the final weight for the Control group, all three OxBC group final weights were higher than the final weights for both the Control and AB groups, and the final weights for the 4 ppm and 8 ppm treatment groups were higher than the final weight for the 2 ppm OxBC group (quadratic, *p* < 0.001). OxBC linearly improved the overall Gain/Feed ratio (G/F) compared to the control, with the 4 and 8 ppm groups also outperforming the AB group.

The effect of OxBC on growth performance was most apparent during the starter period (Study Days 1–28), where all three levels of OxBC produced significant improvements in body weight (BW) (quadratic, *p* = 0.004), average daily weight gain (ADG) (quadratic, *p* = 0.003), average daily feed intake (ADFI) (quadratic, *p* = 0.02), and G/F (quadratic, *p* = 0.001) compared to the Control. Animals in the OxBC groups also performed better than or equivalent to those receiving antibiotics during the starter period in BW (quadratic, *p* = 0.004), ADG (quadratic, *p* = 0.003), ADFI (quadratic, *p* = 0.02), and G/F (quadratic, *p* = 0.001). All measured parameters were higher during the Starter period for the AB group compared to the Control animals (at the quadratic significance levels indicated above) except for ADFI, which was equivalent for both groups.

During the Grower period (Days 29–84), animals in all three OxBC groups continued to show significantly improved BW (linear, *p* < 0.001), ADG (quadratic, *p* = 0.01), and G/F (linear, *p* < 0.001)) relative to the Control. The ADG of the AB group was also higher than that of the Control group (*p* < 0.001), but there were no differences between the OxBC groups and the AB group. There was an apparent dose-dependent decrease in feed consumption across the three OxBC groups, with the ADFI of the 8 ppm OxBC group being significantly lower than the Control (linear, *p* = 0.004). The G/F for all OxBC groups and the AB group were significantly higher compared to the Control (linear, *p* < 0.001). The 8 ppm OxBC group had the highest G/F, which was significantly higher than the G/F for the Control, AB, and the 2 ppm OxBC groups (linear, *p* < 0.001).

During the Finisher stage (Days 85–140), animals in all three OxBC groups showed significantly improved BW compared to the Control and AB groups (linear, *p* < 0.001). There were no significant differences in the ADGs of animals among the five groups, although the higher ADGs of the three OxBC groups approached linear significance relative to both the Control and AB groups (*p* = 0.052). Supplementation with all three levels of OxBC resulted in significant improvements in G/F in comparison to the Control and AB groups (linear, *p* < 0.001).

As was observed in the Grower period, animals receiving OxBC had lower feed intake during the Finisher phase when compared to the Control (linear, *p* < 0.001). The feed intake was significantly lower in the 4 ppm and 8 ppm OxBC groups compared to the Control group (linear, *p* < 0.001), and feed intake in the 8-ppm group was also lower than intake in the AB group (linear, *p* < 0.001). Note that during the Finisher stage, animals in the AB group did not receive antibiotics, which is a common practice and a legal requirement in many jurisdictions, including in Vietnam

In terms of general health, the results presented in Table 2 show that the highest incidences of both diarrhea and mortality were observed in the control group for all study periods, with the lowest being observed in the OxBC groups. The highest incidence of diarrhea occurred during the Starter period, and OxBC significantly and linearly reduced the diarrhea rate in a dose-dependent manner (*p* < 0.001).

The occurrence of diarrhea during the Grower and Finisher periods was markedly less than during the Starter period. Despite the lower background incidence of diarrhea in these later growth phases, OxBC still continued to provide significant and dose-dependent protection from the condition.

During the Grower phase, the DI was lowest in the 8 ppm group, and this was significantly lower than the DI in all four other groups (linear, *p* < 0.001). The DI for the Control group was significantly higher than the DI observed in the AB, 2 ppm, and 4 ppm OxBC groups (linear, *p* < 0.001). The DI during the Finisher phase was the lowest in the 8 ppm group, and this was significantly less than the DI in all four other groups (linear, *p* < 0.001). The DI in the Control group was the highest, and it decreased linearly as the concentration of OxBC increased (2 ppm to 8 ppm) (*p* < 0.001).

Overall, animals in the Control group had the highest incidence of diarrhea, whereas animals in the 8 ppm OxBC group had the lowest incidence. OxBC significantly reduced the DI relative to both the Control and AB groups in a dose-dependent manner (linear, *p* < 0.001).

Mortality was relatively low overall, with few pigs dying during the study. The lowest number of deaths occurred in the OxBC groups, but differences were not significant between any of the groups.

## 4. Discussion

In general, the health and performance measures for all treatment groups fell within the normal range for Vietnam, with the exception of overall ADG, which was moderately lower than expected. This outcome was likely due to a combination of factors, including the use of mash instead of pelleted feed and the presence of seasonal environmental stressors during the trial. Nonetheless, results from this study demonstrate that dietary supplementation with low parts-per-million levels of OxBC improves the productivity of pigs reared under Vietnamese commercial production conditions compared to AGP-supplemented diets. The inclusion of OxBC in the feed led to improvements in ADG and feed efficiency (G/F) during each of the three stages of the production cycle, which resulted in significantly higher final body weights. Whereas supplementation with 2 ppm OxBC gave results comparable to the antibiotic growth promoters, animals provided with 4 and 8 ppm significantly outperformed the antibiotic-positive control group in all of the measured performance and health aspects. The observation that OxBC outperformed the antibiotics underscores observations from previous studies in poultry [6], piglets [5], sows [4], and dairy cows [7] that the benefits of OxBC are most evident in animals experiencing conditions involving some level of stress.

The observation that the benefits of OxBC on performance were most apparent during the Starter period compared to the Grower and Finisher periods is not surprising, given that OxBC also had a significant impact on health during the Starter period. The highest incidence rates of diarrhea occurred during the Starter period, consistent with the well-known susceptibility of young piglets to post-wean diarrhea [13]. The reduced incidence of diarrhea observed in the OxBC-supplemented groups relative to the Control is consistent with the immune supporting actions of OxBC, which center upon an ability to increase the level of pathogen pattern recognition receptors (PPRR) and to enhance the down-stream innate immune response to receptor activation [1,2]. Results from an infectious challenge study in broilers, where dietary supplementation with OxBC reduced *Clostridium perfringens* levels by 2 to 3 log units, are consistent with the concept that OxBC supports the intestinal immune system and allows the host to resist pathogen colonization [6].

The reduction in the incidence of diarrhea likely explains, in part, the improved growth performance of piglets in the OxBC-supplemented versus the Control group during the Starter period. As would be expected for older, more immunocompetent pigs, there were fewer incidences of diarrhea in Grower and Finisher pigs compared to Starter pigs across all treatment groups.

OxBC is composed of a mixture of immunologically active polymer compounds, which represent the bulk of the product, in combination with minor amounts of many low molecular weight apocarotenoid compounds [1,14]. Several of the apocarotenoids present in OxBC are recognized flavor agents [14,15]. Thus, in addition to immune-supporting activity, OxBC may also improve feed palatability. Feed intake is known to be a critical factor for the health and performance of pigs and is of particular importance for post-wean piglets [16]. Improved feed palatability and increased ADFI may represent an additional mode of action that contributed to the improved growth of piglets in the OxBC groups during the Starter period. However, feed intake was stable across all treatments during the Grower period, and it decreased in the OxBC groups during the Finisher period. This decrease occurred starting at 2 ppm OxBC versus the Control group and at 4 ppm OxBC versus the AB group. As well, feed intake was lower in pigs fed 4 and 8 ppm OxBC versus the Control group for the entire 140-day growth period and in pigs fed 8 ppm OxBC versus the AB group for 140-days. Rather than stimulating feed intake, the post-wean effect beyond 25 days may have been to improve the health of the animals and thereby allow the more efficient utilization of nutrients for growth promotion in lieu of supporting immune activities. This may have resulted in a nutrient (calorie)-sparing response, which would be contrary to the more common observation of reduced feed intake in the presence of an immune challenge [16].

Overall, the results of this study demonstrate the benefits of dietary supplementation with OxBC on growth performance, feed efficiency, and the health of post-weaned piglets. These findings have significance for both the commercial feed industry and the field of nutrition. For the feed industry, the findings support the application of OxBC as an effective alternative to antibiotic growth promoters. The performance and health benefits obtained with the 4 and 8 ppm OxBC groups were significantly better than those of the antibiotic growth promoter positive control group.

## 5. Conclusions

OxBC improved the growth rate, feed efficiency, and body weight of pigs compared to unsupplemented control animals in a manner similar to antibiotics, with pigs that received 4 and 8 ppm OxBC performing better than the animals that received antibiotics. In Starter pigs, OxBC reduced the incidence of diarrhea in a dose-dependent manner (2, 4, and 8 ppm) and to a greater extent than did antibiotics. These findings support the concept that oxidized β-carotene can enhance swine growth and health under tropical husbandry conditions, and they further validate the effectiveness of this strategy in enhancing animal performance in the absence of in-feed antibiotics.

## Figures and Tables

**Table 1 animals-12-03200-t001:** Composition of basal diets for Starter, Grower, and Finisher phases ^1^.

Composition (g/kg)				Nutrient Composition			
Ingredient	Post-Wean	Grower	Finisher	Ingredient	Post-Wean	Grower	Finisher
Corn	481.0	700.00	513.32	Dry matter, %	88.15	87.85	88.04
Rice bran	-	58.08	123.07	ME, Kcal/kg	3200	3265	3265
Cassava root	-	10.00	157.86	Protein, %	22.0	18.0	15.5
Soybean meal 47.5% CP	237.0	136.92	100.00	Fat, %	6.23	5.84	7.03
Whey powder 11% CP	100.0	-	-	Fiber, %	2.34	2.41	2.77
Prelac 38% CP	100.0	-	-	Calcium, %	0.90	0.60	0.60
Fish meal 50% CP	-	75.81	74.00	Total phosphorus, %	0.70	0.57	0.57
Fish meal 60% CP	20.0	-	-	Available phosphorus, %	0.52	0.48	0.48
Soybean oil	27.1	11.83	25.72	Lysine, %	1.42	1.07	0.92
Premix Min-Vit	3.0	2.50	2.50	Methionine, %	0.47	0.39	0.32
DCP	18.0	-	-	Methionine + Cysteine, %	0.82	0.69	0.61
Seashell powder	3.4	-	-	Threonine, %	0.91	0.81	0.70
Enzyme	1.0	-	-	Tryptophan, %	0.27	0.21	0.19
Salt	2.9	2.45	2.51				
Sweetener	0.3	-	-				
L-Lysine	3.2	1.57	0.63				
DL-Methionine	1.6	0.26	0.13				
L-Threonine	1.3	0.48	0.26				
DL-Tryptophan	0.2	0.11	0.0				
Total	1000.0	1000.0	1000.0				

^1^ CP, crude protein; DCP, dicalcium phosphate; ME, metabolizable energy.

**Table 2 animals-12-03200-t002:** Effect of oxidized β-carotene (OxBC) on growth performance of pigs from post-weaning to finish ^1,2^.

	Treatments						*p*-Value		
Item	Control ^3^	AB ^3^	OxBC	OxBC	OxBC	SEM ^4^	Combined ^5^	Linear ^5^	Quadratic ^5^
			2 ppm ^3^	4 ppm ^3^	8 ppm ^3^				
BW Day 0 (kg)	7.66 ± 0.13	7.78 ± 0.08	7.62 ± 0.15	7.66 ± 0.11	7.78 ± 0.08	0.03	0.12	0.47	0.31
**Days 0 to 28**									
BW Day 28 (kg)	17.7 ± 0.3 ^c^	18.4 ± 0.3 ^b^	19.8 ± 0.5 ^a^	20.0 ± 0.5 ^a^	20.2 ± 0.5 ^a^	0.2	<0.001	<0.001	0.004
ADG (g)	359 ± 8 ^c^	381 ± 12 ^b^	435 ± 20 ^a^	440 ± 19 ^a^	442 ± 15 ^a^	8	<0.001	<0.001	0.003
ADFI (g)	601 ± 10 ^b^	607 ± 20 ^b^	689 ± 29 ^a^	676 ± 27 ^a^	684 ± 21 ^a^	9	<0.001	0.002	0.02
Gain/Feed	0.59 ± 0.11 ^c^	0.63 ± 0.01 ^b^	0.63 ± 0.01 ^b^	0.65 ± 0.01 ^a^	0.65 ± 0.01 ^a^	0.01	<0.001	<0.001	0.001
DI (%)	7.93 ± 0.56 ^a^	6.60 ± 0.52 ^b^	5.96 ± 0.21 ^c^	4.86 ± 0.36 ^d^	3.68 ± 0.60 ^e^	0.31	<0.001	<0.001	0.83
**Days 29 to 84**									
BW Day 84 (kg)	49.6 ± 0.3 ^c^	51.9 ± 0.8 ^b^	53.2 ± 0.4 ^ab^	53.7 ± 0.5 ^a^	53.8 ± 0.4 ^a^	0.3	<0.001	<0.001	<0.001
ADG (g)	569 ± 2 ^b^	598 ± 17 ^a^	596 ± 5 ^a^	601 ± 14 ^a^	600 ± 11 ^a^	3	0.001	0.001	0.01
ADFI (g)	1645 ± 78 ^a^	1631 ± 74 ^a^	1625 ± 33 ^a^	1590 ± 34 ^ab^	1542 ± 29 ^b^	12	0.047	0.004	0.29
Gain/Feed	0.35 ± 0.02 ^c^	0.36 ± 0.01 ^b^	0.37 ± 0.01 ^b^	0.38 ± 0.01 ^ab^	0.39 ± 0.01 ^a^	0.01	<0.001	<0.001	0.76
DI (%)	2.54 ± 0.55 ^a^	1.97 ± 0.43 ^b^	1.80 ± 0.47 ^bc^	1.46 ± 0.33 ^bc^	1.33 ± 0.24 ^c^	0.12	0.002	<0.001	0.31
**Days 85 to 140**									
BW Day 140 (kg)	99.5 ± 0.5 ^d^	101.7 ± 0.5 ^c^	103.6 ± 0.5 ^b^	104.2 ± 0.2 ^ab^	104.4 ± 0.4 ^a^	0.4	<0.001	<0.001	<0.001
ADG (g)	892 ± 13	889 ± 8	901 ± 9	903 ± 7	904 ± 9	2	0.052	0.007	0.87
ADFI (g)	3085 ± 114 ^a^	3032 ± 101 ^ab^	2959 ± 87 ^bc^	2897 ± 47 ^c^	2876 ± 49 ^c^	22	0.004	<0.001	0.6
Gain/Feed	0.29 ± 0.01 ^b^	0.29 ± 0.01 ^b^	0.31 ± 0.01 ^a^	0.31 ± 0.01 ^a^	0.31 ± 0.01 ^a^	0.002	0.001	<0.001	0.69
DI (%)	0.86 ± 0.06 ^a^	0.70 ± 0.05 ^b^	0.64 ± 0.07 ^b^	0.68 ± 0.05 ^b^	0.55 ± 0.07 ^c^	0.02	<0.001	<0.001	0.11
**Days 0 to 140**									
ADG (g)	656 ± 4 ^d^	671 ± 4 ^c^	686 ± 4 ^b^	690 ± 1 ^ab^	690 ± 3 ^a^	3	<0.001	<0.001	<0.001
ADFI (g)	2019 ± 81 ^a^	1990 ± 71 ^b^	1974 ± 45 ^abc^	1933 ± 28 ^bc^	1907 ± 32 ^c^	13	0.03	0.002	0.82
Gain/Feed	0.32 ± 0.01 ^c^	0.34 ± 0.01 ^b^	0.35 ± 0.01 ^b^	0.36 ± 0.01 ^ab^	0.36 ± 0.01 ^a^	0.003	<0.001	<0.001	0.25
DI (%)	3.03 ± 0.17 ^a^	2.43 ± 0.20 ^b^	2.19 ± 0.15 ^c^	1.85 ± 0.15 ^d^	1.50 ± 0.07 ^e^	0.1	<0.001	<0.001	0.12
Mortality (%)	5.00 ± 3.54	3.00 ± 2.74	2.00 ± 2.74	2.00 ± 2.74	2.00 ± 2.74	0.58	0.43	0.11	0.32

^a–e^ Within a row, means with differing superscripts differ (*p* < 0.05). The linear model used included block (body weight), treatment, and block × treatment to evaluate the difference among the five treatments. ^1^ Values are represented as means ± SD, ^2^ BW, body weight; ADG, average daily gain; ADFI, average daily feed intake; DI, diarrhea incidence, ^3^ Control diet was the basal diet without any supplementation of antibiotics or OxBC (oxidized β-carotene); AB diet contained antibiotics as follows: Pre-wean (nursery): 100 ppm colistin sulfate; Starter (days 0–28): 150 ppm chlortetracycline and 100 ppm colistin sulfate; Grower (days 29–84): 150 ppm chlortetracycline; Finisher (days 85–140): no antibiotics. OxBC 2 ppm, 4 ppm, and 8 ppm diets contained 2 mg/kg, 4 mg/kg, and 8 mg/kg OxBC, respectively, ^4^ SEM is for treatment within the same row, ^5^ Combined, linear and quadratic *p*-values were determined for treatment within the same row.

## Data Availability

Not applicable.

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
