# Peer review of "Effect of Oxidized β-Carotene on Swine Growth Performance under Commercial Production Conditions in Vietnam"

_animals, 2022, doi:10.3390/ani12223200_

Round 1
Reviewer 1 Report
The manuscript describes a very simple experiment conducted on 500 pigs during the Starter, Grower and Finisher periods.
In the methodology of the tests, there is no information on the control of the health of pigs (daily?), the assessment of feces for diarrhea, poor results in fattening, because daily weight gains below 700 grams (650-690). The causes of diarrhea (culture?) have not been studied.
The low scientific level of the work means that – in my opinion – the manuscript is not suitable for the journal Animals.
Author Response
See Reviewer 1.docx file

Reviewer 2 Report
Comments to the Authors of manuscript number: animals-1982765 entitled “Effect of Oxidized β-Carotene on Swine Growth Performance Under Commercial Production Conditions in Vietnam”.
The study was performed on pigs both genders for the whole production cycle. It is very interesting study, however there are doubts relating the diet. It is not clear if the OxBC was given on top or not? And how in this case the diet was balanced?
1. L 77-80 the reference should be added
2. The clear hypothesis should be added before the goal of the study
3. The study design is presented very clearly
4. The distribution and replication is well done
5. It is not clear if carotene was given on top or was included in the basal diet? Especially that it was given at different concentration. This diet was not balanced.
Author Response
See Reviewer 2.docx file

Reviewer 3 Report
The manuscript is interesting, the experiment design is suitable and the methods are well selected. I have some minor suggestions:
In the introduction: it might be interesting if the authors could add some information about pig production in Vietnam and the antibiotic uses as well as the current status of finding an alternative for antibiotics.
Why did the authors choose the commercial farm in the study?
Do these animals in the current study has any family relationship (such as having the same dams, sires, or grandparents)?
Line 147-154: Did the authors perform the posthoc test?
Line 149-150: Please justify why choosing the quadratic model for determining the dose-dependent effect.
Author Response
See Reviewer 3.docx file

Round 2
Reviewer 1 Report
Dear Sirs,
I maintain my position that the presented manuscript presents too low a scientific level. The Authors, despite the corrections made, will not change the results of fattening. Daily body gain from 656 to 690 g per head should be considered weak. It is true that the Authors supplemented the text with a control of the health status of pigs in the experiment, but nevertheless, in my opinion, performing such a simple experiment is not enough.
